# Circulating Tumour Cells: Detection and Application in Advanced Non-Small Cell Lung Cancer

**DOI:** 10.3390/ijms242216085

**Published:** 2023-11-08

**Authors:** Kalliopi Andrikou, Tania Rossi, Alberto Verlicchi, Ilaria Priano, Paola Cravero, Marco Angelo Burgio, Lucio Crinò, Sara Bandini, Paola Ulivi, Angelo Delmonte

**Affiliations:** 1Medical Oncology Department, IRCCS Istituto Romagnolo per lo Studio dei Tumori (IRST) “Dino Amadori”, 47014 Meldola, Italy; kalliopi.andrikou@irst.emr.it (K.A.); alberto.verlicchi@irst.emr.it (A.V.); ilaria.priano@irst.emr.it (I.P.); paola.cravero@irst.emr.it (P.C.); marco.burgio@irst.emr.it (M.A.B.); lucio.crino@irst.emr.it (L.C.); angelo.delmonte@irst.emr.it (A.D.); 2Biosciences Laboratory, IRCCS Istituto Romagnolo per lo Studio dei Tumori (IRST) “Dino Amadori”, 47014 Meldola, Italy; sara.bandini2@irst.emr.it (S.B.); paola.ulivi@irst.emr.it (P.U.)

**Keywords:** CTCs, epithelial-to-mesenchymal transition (EMT), lung cancer, NSCLC, prognosis, treatment

## Abstract

Non-small cell lung cancer (NSCLC) is one of the deadliest diseases worldwide. Tissue biopsy is the current gold standard for the diagnosis and molecular profiling of NSCLC. However, this approach presents some limitations due to inadequate tissue sampling, and intra- and intertumour heterogenicity. Liquid biopsy is a noninvasive method to determine cancer-related biomarkers in peripheral blood, and can be repeated at multiple timepoints. One of the most studied approaches to liquid biopsies is represented by circulating tumour cells (CTCs). Several studies have evaluated the prognostic and predictive role of CTCs in advanced NSCLC. Despite the limitations of these studies, the results of the majority of studies seem to be concordant regarding the correlation between high CTC count and poor prognosis in patients with NSCLC. Similarly, the decrease of CTC count during treatment may represent an important predictive marker of sensitivity to therapy in advanced NSCLC. Furthermore, molecular characterization of CTCs can be used to provide information on tumour biology, and on the mechanisms involved in resistance to targeted treatment. This review will discuss the current status of the clinical utility of CTCs in patients with advanced NSCLC, highlighting their potential application to prognosis and to treatment decision making.

## 1. Introduction

Lung cancer remains the leading cause of cancer-related deaths worldwide [1]. Non-small cell lung cancer (NSCLC) is a heterogeneous disease that represents about 85% of all lung malignancies [2]. More than 60% of NSCLC patients are diagnosed with advanced disease, and the 5-year survival rate is lower than 15% [1,3]. In the past twenty years, there has been a great improvement in understanding the molecular biology of oncogene-driven tumours, leading to significant advances in the treatment of these patients [4]. At the same time, the treatment of advanced non-oncogene addicted NSCLC has been revolutionized with the introduction of immune checkpoint inhibitors (ICIs) in clinical practice [5]. Unfortunately, not all patients respond, and only a subgroup of them benefit from immunotherapy. Hence, in the era of personalized medicine, there is an urgent need for new predictive biomarkers that may allow early evaluation of treatment response, selection of patients, and identification of drug-resistance.

Based on the current evidence, histopathological analysis and molecular profiling of all NSCLC tissue specimens is necessary before any therapeutic decision-making [6]. However, up to 30% of NSCLC patients do not undergo molecular profiling because of limited quantity or quality of tissue available [7], representing these two features as a strong limitation. Moreover, tissue biopsy is invasive and has low repeatability, thus limiting the possibility to define the intra- and intertumour heterogenicity in both spatial and temporal terms.

Liquid biopsy is a non-invasive method that may identify cancer-related biomarkers in peripheral blood, and can be repeated at multiple timepoints with the potential to define temporal and spatial tumour heterogeneity [8]. Circulating tumour cells (CTCs), due to recent advances in molecular knowledge, represent one of the most studied components of liquid biopsy as prognostic and predictive biomarkers in the management of NSCLC patients.

In this review, a search was conducted on the Pubmed database with the following strategy: “advanced non-small cell lung cancer” and “CTCs” in title in the last 10 years to describe the concept of CTCs and their clinical significance in the management of advanced NSCLC patients.

## 2. Circulating Tumour Cells

Circulating tumour cells (CTCs) are rare; one CTC per 10^6^–10^7^ white blood cells, found in the bloodstream of patients with solid tumours [9]. CTCs serve as a “liquid biopsy”, together with circulating metabolites, extracellular vesicles (EVs), and circulating nucleic acids such as cell-free DNA (cfDNA), microRNAs, and messenger RNA (mRNA) [9]. Liquid biopsy refers to the cytological and molecular analysis of cancer markers shed by the tumour into blood. It represents a minimally invasive and easily repeatable test, allowing for longitudinal studies. Collection and isolation of CTCs is more complex compared to ctDNA or cfDNA, but it allows protein, transcriptomic, or methylation analyses. In Figure 1, we report an overview of liquid biopsy approaches, including advantages and disadvantages (Figure 1).

CTCs slough off of the primary tumour and, through a process known as intravasation, enter the circulating blood [10], initiating the metastatic process. They can migrate either as single cells (single CTCs), or as clusters of two or more CTCs (CTC clusters or circulating microemboli), held together by intercellular junctions [11,12]. Single CTCs and CTC clusters have a short half-life in the bloodstream (6–10 min for CTC clusters and 25–30 for single CTCs) [13], as they face several insults such as attack by immune cells, cell death by anoikis, and physical stress due to fluid shear, limiting the metastatic diffusion. However, CTC clusters are associated with poor clinical outcome in cancer patients, including lung cancer, and are characterized by a significant higher metastatic potential thanks to their stem features [11,14]. CTC clusters can be further distinguished in homotypic CTC clusters, composed of cancer cells only, and heterotypic CTC clusters, formed by cancer cells and other kind of cells, including nonmalignant immune and stromal cells [12]. Clustering of CTCs with neutrophils has been widely studied, as this interaction led to cell cycle progression in circulation and metastatic seeding acceleration [15,16]. The potential of CTCs as metastatic-initiating cells was proven in xenograft models [17], while in patients, a higher number of CTCs is correlated with increased tumour aggressiveness and decreased time to relapse in different cancer types, including NSCLC [18,19,20,21].

Therefore, due to the crucial role of CTCs in the metastatic cascade, many researchers support CTC investigation, as a tool to obtain further information concerning the tumour phenotype and genotype, and to predict disease progression and survival in early and advanced stage cancer patients.

### 2.1. Methods for CTC Isolation and Detection in NSCLC

As reported above, CTCs are rare, with a limited survival time in the bloodstream, making both their identification and isolation difficult and challenging [12]. Various methods have been developed for CTC enrichment and isolation, which can be performed either by positive selection [22,23] or negative selection [24,25,26], as reported in Table 1.

As the origin of most cancers is epithelial, several enrichment methods have been developed to select cells with positive expression of the epithelial cell adhesion molecule (EpCAM) antigen [34]. In this context, one of the most used platforms for CTC immune-magnetic enrichment, isolation, and enumeration relies on the EpCAM-dependent method CellSearch^®^ technology (Veridex, Raritan, NJ). More specifically, CellSearch-based CTC enumeration is a helpful tool to stratify patients into favourable and unfavourable prognostic groups in advanced breast, colon, and prostate tumours [35,36,37,38], and represents the only FDA-approved CTC-based assay [39]. Concerning advanced NSCLC patients, the CellSearch detection of >5 CTCs/7.5 mL of blood was considered as a poor prognostic factor. However, CTC are not detectable in two-thirds of stage IV NSCLC patients, while in stage III NSCLC patients CTCs are detectable in less than 5% of patients [27,28]. Other EpCAM-dependent CTC detection methods based on immuno-affinity include the Adnatest assay (Qiagen) and the GILUPI Cell-collector [40].

However, the use of EpCAM as the main CTC marker may limit their detection, due to the variable expression of different markers. For instance, in the case of NSCLC using label-independent procedures, a higher number of CTCs were detected than using EpCAM-dependent methods, consistent with the low CTC-positivity rates at CellSearch [27,41]. The variable marker expression is due to epithelial–mesenchymal transition (EMT), which plays a significant role in the spread of systemic cancer. During EMT, cancer cells lose their epithelial features by a downregulation of epithelial genes, such as E-cadherin, EpCAM, occludins, claudins, and α- and β-catenin [42,43]. On the other hand, they acquire mesenchymal properties by an upregulation of mesenchymal genes such as N-cadherin, matrix metalloproteinases, integrins α_v_ and β_1_, vimentin, and fibronectin [42,43]. In addition, some transcriptional factors such as Snail and Twist have a crucial role in survival of CTCs during EMT process [43]. By combining epithelial (EpCAM and CK8/18/19) and mesenchymal (vimentin and TWIST1) markers, three categories of CTCs were identified: epithelial CTCs (E-CTCs), mesenchymal CTCs (M-CTCs), and hybrid epithelial/mesenchymal phenotypes (E/M-CTCs) [43,44]. While E/M-CTCs were shown to be predictive in distinguishing patients with malignant NSCLC from those with benign disease, M-CTCs were associated with the presence of metastasis [43,44,45]. Interestingly, CTCs most frequently maintain a hybrid E/M phenotype, allowing them to retain their epithelial properties, but with an increase of the aggressive potential due to mesenchymal features [46,47].

Several studies demonstrated that all of these changes confer a higher metastatic potential and chemoresistance to cancer cells. Zhang et al. demonstrated the association between the EMT phenotype of CTCs from the peripheral blood and distant metastasis in patients with NSCLC [45]. A total of 110 patients were evaluated: 85 patients had NSCLC, of which 41 were with metastatic and 44 were with nonmetastatic disease, and 25 were with benign diseases. Overall, 80% of patients were characterized as CTC-positive with EMT markers (≥ 1 cell/5 mL blood) using a Canpatrol™ CTC assay; the CTC-positive rate was not related to the nonmetastatic and metastatic status. The total number and the mean numbers of CTCs of each subpopulation were statistically higher in NSCLC than in patients with benign pulmonary diseases. In particular, E/M- and M-CTCs rates were significantly superior in patients with NSCLC (75.3 and 44.7%, respectively) than in those with benign pulmonary diseases (0% and 0%, respectively; *p* < 0.05). Finally, M-CTCs were significantly higher in metastatic than in nonmetastatic patients (*p* < 0.001) [45]. Given these data, the EpCAM-negative CTC population may be not recognized, and detection approaches based on CTC epithelial markers may fail to detect those with mesenchymal characteristics [48].

Among the marker-independent approaches, isolation by size of epithelial tumour cells (ISET) is another method which can isolate CTCs [49]. In a study by Krebs et al., conducted on chemo-naive patients with stage IIIA to IV NSCLC, the CellSearch platform detected CTCs in 9 patients (40%), and while using ISET, 32 patients (80%) emerged as CTC-positive in a subpopulation of CTCs negative for the expression of epithelial markers [27]. Moreover, the CellSearch platform did not detect circulating tumour microemboli, while the ISET system detected them in 43% of patients [27]. However, this approach could underestimate very small CTCs [9]. In addition to their size, deformability is another property that can be exploited for CTC detection through the Parsortix microfluidic platform, which allows the recovery of CTCs and CTC clusters with preserved viability [50]. In NSCLC, the Parsortix platform provided the highest CTC-positivity compared to other technologies such as ISET and Ficoll. In addition, CTC detection using Parsortix was associated with disease progression, reduced PFS, and a high risk of relapse in NSCLC patients treated with anti-PD-1 agents [51]. In addition, microfluidic devices allow the continued collection of CTCs for further downstream analysis, and offer the possibility to integrate both isolation and detection of CTCs in a single device [52].

Other emerging approaches have been described to be suitable for CTC detection, such as spectroscopy-based techniques. Surface-enhanced Raman spectroscopy (SERS) allows for the highly sensitive and accurate detection of single cells by generating molecular fingerprint signals [53]. This method was found to be rapid and cost-effective, with high recognition rates (90%), and bio-probes with specific aptamers on parylene micropore membranes could distinguish cancer cells from white blood cells [54]. Recent findings have demonstrated that SERS can be functionalised and complemented with machine learning to detect metastasis-initiating cells directly in the blood of lung cancer patients, with a 100% diagnostic sensitivity [55].

### 2.2. Molecular Evaluation of CTCs

#### 2.2.1. Single Cell Analysis

Thanks to the efforts made in technology development, molecular evaluation of CTCs is now possible, even at the single-cell level, allowing for the genomic and transcriptomic characterization of individual CTCs [56].

Concerning single-cell genomics of CTCs, many studies have focused on the assessment of single nucleotide variants (SNVs). The first technologies available for this application consisted of polymerase chain reaction (PCR) genotyping, especially digital PCR (dPCR). This low-input technique allows highly sensitive and specific detection of mutations; indeed, dPCR targets specific sequences due to the presence of primers, which can be combined to allow target multiplexing. dPCR-based SNV detection at the single-cell level has been demonstrated to be a useful tool for monitoring patients throughout the course of a disease, in addition to unveiling their genetic heterogeneity [57,58,59]. In NSCLC, the mutational analysis of epidermal growth factor receptor (*EGFR*) through dPCR on single CTCs isolated prior treatment with tyrosine kinase inhibitors (TKIs), identified a low mutation rate at the single-CTC level in patients positive for *EGFR* mutations. At the same time this approach can detect rare *EGFR* mutations in those patients considered as wild-type [60]. However, the initial limitation of these techniques was the fact that they were able to detect only specific alterations, limiting the discovery of other variants. More recently, the development of panels for next generation sequencing (NGS), including oncology-relevant genes, was implemented. Pailler et al. investigated resistance mutations in 48 cancer-related genes in single CTCs from ALK-rearranged NSCLC patients treated with crizotinib or lorlatinib at disease progression. In crizotinib-treated patients, mutational heterogeneity in multiple genes involved in ALK-independent pathways (“off-target” mutations), such as the RTK-KRAS (*EGFR*, *KRAS*, *BRAF* genes) and *TP53* pathways, were identified. CTCs from lorlatinib-treated patients harboured mutations affecting the *ALK* gene, with some variants not detected in the corresponding tumour biopsy [61]. Single-cell whole exome sequencing was also demonstrated to be critical for studying drug resistance in NSCLC, as well as to discover new drug targets in NSCLC. Chang et al. exploited this approach to explore the mutational landscape of CTCs compared to matched primary and progressive tumours in platinum-based treated NSCLC patients. In addition to the frequency of mutations in cancer-driver genes (i.e., *EGFR* and *TP53*), genes associated with cell-cycle and stem cell features were frequently altered in CTCs, most of which were derived from primary tumour samples, and played crucial roles in chemo-drug resistance and metastasis for NSCLCs [62].

Transcriptomic analysis of single CTCs through RNA sequencing is a technology that may help to better understand the molecular mechanisms underlying metastasis [63], although few studies are present in the literature for NSCLC. One of the most recent technologies relies on the single-cell 3′ RNA sequencing for gene expression analysis using the 10X Chromium platform [64]. In CTCs isolated from diagnostic leukapheresis of NSCLC patients, single-cell 10X Chromium was able to identify heterogeneous CTC phenotypes. The epithelial-like phenotype was characterized by the expression of epithelial markers, as well as Ki67 (highly proliferation) and IL-1B, and interferon-response pathways (immune responsive). The mesenchymal/invasive phenotype showed expression of vimentin, hypoxia, and glycolysis pathways, whereas CTCs from the mesenchymal/stem cell-like phenotype were enriched in genes including *ALDH13*, and genes associated with adipogenesis [65].

#### 2.2.2. PD-L1 Determination of CTCs

Several studies have been conducted in order to evaluate the expression of the programmed death-ligand 1 (PD-L1) in CTCs from patients treated with immune checkpoint inhibitors, especially with immunofluorescence techniques. In advanced NSCLC patients, it has been shown that CTCs were more frequently PD-L1-positive compared to tumour tissue, therefore representing a potential tool to identify patients who would benefit from immunotherapy [66,67]. In 96 NSCLC patients treated with nivolumab, baseline PD-L1 immunofluorescent expression in CTCs (83%) was higher than tissue (41%), and a higher number of PD-L1 positive CTCs was associated with nonresponders (PFS < 6 months) (*p* = 0.04). All of the patients had PD-L1 positive CTCs at disease progression [67]. Similar data concerning the increased PD-L1 positivity in CTCs compared to tumour tissue (53% and 43.2%, respectively) in advanced NSCLC patients were reported by Zhou et al. [66].

Several techniques have been described for the assessment of PD-L1 in CTCs. Among them, immunofluorescence methods were described, with the use of fluorescent antibodies targeting PD-L1, following CTC enrichment [66,67,68]. Other studies have proposed the CellSearch system, by including an additional antibody targeting PD-L1 in the empty fluorescence channel (AlexaFluor488 or fluorescein) [69,70]. Other techniques for PD-L1 expression assessment in CTCs include the DEPArray platform (Menarini Silicon Biosystems). This technology allows one to visually inspect single cells marked with antibodies targeting specific antigens (i.e., PD-L1), and to recover the cells of interest at a single-cell resolution by exploiting the dielectrophoretic principle [71]. One of the main advantages of the DEPArray technology relies on the possibility to analyse the expression of PD-L1 at single-cell resolution, as already reported in advanced urothelial carcinoma [72].

## 3. Prognostic Value of CTCs in Advanced NSCLC

The prognostic role of CTC count, evaluated at the baseline and after one cycle of chemotherapy by a semiautomated EpCAM–based immunomagnetic technique, has been evaluated for the first time in 101 chemotherapy-naive patients with stage III and IV NSCLC [28]. The number of CTCs in 7.5 mL of blood was higher in 60 patients with stage IV NSCLC (range, 0 to 146) than in 27 patients with stage IIIB (range, 0 to 3); in 14 IIIA patients, CTCs were not detected. In a univariate analysis, progression-free survival (PFS) was 6.8 vs. 2.4 months (*p* = 0.001), and overall survival (OS) was 8.1 vs. 4.3 months (*p* = 0.001) for patients with <5 CTCs per 7.5 mL of blood compared with ≥5 CTCs before chemotherapy, respectively. In a multivariate analysis, CTC number at baseline was the strongest predictor of OS, with a hazard ratio (HR) of 7.92 (95% CI, 2.85 to 22.01; *p* = 0.001) [28]. Similar results have been seen after a cycle of chemotherapy. Similarly, Nieva et al. evaluated a method using enrichment-free fluorescent labeling of CTCs, followed by automated digital microscopy in 28 patients with NSCLC [73]. CTCs were identified in 68% of analysed samples, with a median concentration of 1.6 per ml (range 0–182.6 CTCs/mL). Increased CTC detection seems to be related to progressive disease in individual patients; patients were grouped into those with ≥ 5 CTCs/mL and those with <5: the first group had a median survival of 244 days, while the second did not reach the median survival, with a median follow-up of 304 days. The HR for death was 4.0 in the group, with ≥5 CTCs/mL relative to those patients with a lower count, with a four-fold risk of dying at any given timepoint for patients with higher CTC counts (*p* = 0.0084) [73]. In another series of 43 advanced NSCLC, blood samples were obtained at the baseline, before the second and fifth cycles of chemotherapy. At baseline, 18 (41.9%) patients were positive for intact CTC counts, and 10 (23.2%) of them had ≥5 CTCs. The group of patients with CTC > 5 at baseline presented worse PFS and OS than those with <5 CTC (*p* = 0.034 and *p* = 0.008, respectively); during therapy, patients with a high amount of CTCs during the treatment demonstrated lower OS and PFS rates [74]. Moreover, they demonstrated that PFS and OS were shorter in patients with a progressive increase of CTC count from the baseline to final timepoint (*p* = 0.003 and *p* = 0.019, respectively) [74]. The prognostic role of CTCs has been evaluated in a pooled analysis of data from 550 patients with NSCLC enrolled in seven European centres. CTC counts of ≥ 2 and ≥ 5 per 7.5 mL were associated with reduced PFS (≥2 CTCs: HR = 1.72, *p* < 0.001; ≥ 5 CTCs: HR = 2.21, *p* < 0.001) and OS (≥2 CTCs: HR = 2.18, *p* < 0.001; ≥ 5 CTCs: HR = 2.75, *p* < 0.001), respectively [75]. In general, clinical trials performed with CellSearch methodology highlight the prognostic value of CTC count, even if negative studies have been reported in the literature [76,77]. Several methodologies, other than CellSearch, have been evaluated, including flow cytometry, the antigen-based platform Cytell, and ISET [49,78,79].

Despite interesting insights suggesting that these alternative methods may have higher sensitivity, the CellSearch approach remains the only clinically validated, FDA-cleared system for identification, isolation, and CTC counts.

Furthermore, interesting results regarding the prognostic role of CTCs in different molecular subtypes of advanced NSCLC have been reported in some studies. Tong et al. investigated the prognostic value of CTC count in 43 patients with *EGFR* -mutated or *ALK* -rearranged NSCLC at baseline and at progression of disease. Increased CTC count at baseline (≥8 CTCs/3.2 mL) was significantly associated with both shorter PFS and OS (11.6 vs. 8.5 months, *p* = 0.004 for PFS and 21 vs. 17.7 months, *p* = 0.013 for OS) [80]. Similar results were reported regarding the prognostic role of CTCs in terms of PFS in NSCLC patients treated with osimertinib [81]. According to CTC values at the baseline, patients were divided into favourable (CTCs < 5) and unfavourable (CTCs ≥ 5) prognostic groups; PFS was 9.3 in the former group and 6.5 months in the later one (HR 5.712, 95% CI: 3.781–9.577; *p* = 0.0002) [81]. The negative prognostic role of CTC count was also reported in a small cohort comprising 24 metastatic *EGFR* -positive NSCLC patients [58]. Patients with CTCs ≥ 1 in 7.5 mL had significantly shorted OS compared to patients without CTCs (3.0 months vs. not reached, *p* = 0.012, HR = 2.9, 95% CI 1.6–54.1 months) [82].

In Table 2, we report some recent studies evaluating the prognostic role of CTCs in advanced NSCLC.

**Table 2 ijms-24-16085-t002:** Recent studies evaluating the prognostic role of CTCs in advanced NSCLC. Abbreviations: NSCLC, non-small cell lung cancer; CTC, circulating cancer cell; chemo, chemotherapy; FISH, fluorescence in situ hybridization; PFS, progression-free survival; OS, overall survival; ISET, isolation by size of tumour cells; PD-L1, programmed cell death ligand 1; WBCs, white blood cells; PD-1, programmed cell death protein 1; NA, not applicable; vs, versus; EML4, echinoderm microtubule-associated protein-like 4; *ALK*, anaplastic lymphoma kinase; PD, progressive disease; Nivo, nivolumab; TKI, tyrosine kinase inhibitor; *EGFR*, epidermal growth factor receptor; RR, response rate; mo., months.

CTC Biomarkers	Study	Analysis Time-Point	Patients (n)	Detection Method	Therapeutic Approach	Main Findings
NA	J Li et al. [83]	At baseline, before 2 and 4 cycles of chemo, or after 1 and 2 mo. of targeted therapy	100	Negative enrichment of immune magnetic beads and FISH	Chemo /*EGFR* TKIs	Patients were divided into low CTC levels (<4 CTCs, LL) and high CTC levels (≥4 CTCs, HL). PFS: 5.6 mo. in LL group vs. 4.2 mo. in HL group, (*p* < 0.001).
	Alama et al. [84]	At baseline, after 4 and 8 cycles	89	ScreenCell CYTO	Nivolumab	OS in patients with ≤2 CTCs was 8.8 mo. vs. 6.2 mo. in patients with ≥2 CTCs (*p* = 0.05).
	Zhou et al. [85]	At the 1st and the 3rd cycle	59	CellSearch	Chemo	Pts with CTC ≥ 2 had a significant shorter PFS and OS (4.3 vs. 4.9 mo. and 8.3 vs. 11.2 mo., respectively)
PD-L1/PD-1	Ilié et al. [86]	At baseline	106	ISET^®^ platform; Rarecells	Chemo	Trend for worse OS in patients receiving first-line cisplatin-based chemotherapy, whose tumours express PD-L1 in CTCs or immune cells
	Kallergi et al. [87]	At baseline and after 3 cycles	30	ISET^®^ platform; Rarecells	Chemo	CTCs were detected in 93.3% and 81.8% of patients at baseline, and after the third chemotherapy cycle, respectively. The presence of >3 PD-1 + CTCs before treatment is associated with shorter PFS (0.5 vs. 3.9 months, *p* = 0.022)
	Cheng et al. [88]	At baseline	66	ISET^®^ platform	NA	PFS time of initial treated patients with positive PD-L1 expression was shorter than that of those with negative PD-L1 expression in CTCs or tumour tissue (*p* > 0.05).
	Sinoquet et al. [89]	At baseline, or later, at progression	54	CellSearch	NA	CTCs and PD-L1(+) CTCs were detected in 43.4% and 9.4% of patients with NSCLC, respectively. PD-L1 expression concordance between tumour tissue and CTCs was low (54%). CTCs and PD-L1(+) CTCs were associated with worse OS, whereas PD-L1 expression in tumour tissue was not. Survival was worse in patients with PD-L1(+) CTCs
*EGFR*	Yang et al. [81]	At baseline and at day 28	68	CellSearch	Osimertinib	Patients in the favourable group (<5 CTCs) at baseline exhibited significantly longer PFS compared with patients in the unfavourable group (≥5 CTCs) (9.3 vs. 6.5 mo.; *p* = 0.0002).
*ALK*	Rossi et al. [90]	At baseline, at the end of first cycle of therapy (T1), and at the first and subsequent radiological assessments and/or at PD	199	CellSearch Expanded cytokeratins profile (EA)	NA	Pts with ≥4 CTCs had a significant lower PFS (0.29 vs. 0.66 years; *p* = 0.004) and OS (0.59 vs. 1.29 years, *p* = 0.04). Similar results using 5 CTCs as cut-off value. EML4-ALK(+) CTCs were associated with shorter PFS compared to NSCLC that did not express EML4-ALK in CTCs (0.57 years vs. 0.94 years, *p* = 0.017).
PD-L1, *ALK*, *EGFR*	Kulasinghe et al. [91]	At baseline	33	ClearCell FX	Nivo Chemo TKI	PFS was not found to be associated with CTCs prior to therapy (*p* = 0.0632), nor the presence of PD-L1 expression (*p* = 0.4023).
11 clinically relevant genes, including *EGFR* and *ALK*	Tamminga et al. [92]	At baseline	86	CellSearch	Chemo TKI	RR of patients with CTC were lower than in patients without CTC (OR = 0.22, *p* < 0.01). In both treatment groups, the difference in RR between patients with and without CTC was similar (interaction *p* = 0.17). No significant interaction between CTC presence and therapy was observed (*p* = 0.42 for PFS and *p* = 0.83 for OS).

## 4. Predictive Role and Clinical Implications of CTCs in Advanced NSCLC

Several studies have investigated the potential predictive role of CTCs as a biomarker during different treatment strategies in advanced NSCLC patients. Specifically, variations of CTC counts and dynamic evolution of actionable genomic mutations during therapy have been investigated as biomarkers for treatment response monitoring. Moreover, interest regarding the role of PD-L1 expression on CTCs has increased in recent years.

### 4.1. Predictive Role during Chemotherapy

Several studies investigated the role of CTC count as a potential biomarker predicting chemotherapy response in advanced NSCLC patients using the CellSearch System (Table 3).

As previously mentioned, in the pivotal clinical trial by Krebs and colleagues, using a positive CTC cut-off a value of ≥2 in 7.5 mL, patients with a reduction of CTC numbers after the first cycle of treatment had a significantly longer PFS (5.4 versus 1.9 months; *p* < 0.001) and OS (8.3 versus 3.3 months; *p* < 0.009) compared with those with an increased or stable number of CTCs [28]. In another study, the association between changes in CTC count and treatment outcome in untreated advanced NSCLC patients became statistically significant only after the second cycle of chemotherapy [93]. A multicentre prospective study evaluated dynamic modification of CTCs during first-line platinum-based chemotherapy in 150 advanced NSCLC patients, monitoring their count at cycle 1, at cycle 4 of therapy, and at disease progression: a persistent presence of CTCs during treatment is correlated with shorter PFS (3 vs. 5.7 months) and OS (5.6 vs. 13.1 months) compared to patients negative for CTCs [94]. Nevertheless, other studies failed to reveal the association of CTC counts and treatment response [76,85].

Moreover, alternative methods, other than CellSearch, have been used to evaluate the validity of the predictive value of CTCs in NSCLC patients treated with chemotherapy (Table 3). The research group of Gorges, using the GILUPI CellCollector, showed that the treatment response during therapy was associated with significant decreases in CTC counts (*p* = 0.001) in 50 advanced NSCLC patients [95]. Another study evaluated the Survivin mRNA-CTCs in 78 patients with advanced NSCLC before, after 1 cycle, and 3 cycles of first-line chemotherapy, showing that persistence of Survivin mRNA-positive CTCs after 1 and 3 cycles was significantly associated with shorter PFS and OS [96].

**Table 3 ijms-24-16085-t003:** Selected studies investigating the predictive role of CTCs in advanced NSCLC patients treated with chemotherapy. Abbreviations: NSCLC, non-small cell lung cancer; CTC, circulating cancer cell; Chemo, chemotherapy; Pts, patients; FISH, fluorescence in situ hybridization; PFS, progression-free survival; OS, overall survival; PD, progressive disease; TKI, tyrosine kinase inhibitor; RR, response rate; mo., months; vs., versus.

Study	Analysis Time-Point	Patients (n)	Therapeutic Approach	Detection Method	Predictive Relevance
Krebs et al. [28]	At baseline and after 1st cycle	101	Chemo	CellSearch	Pts with <5 CTCs count at the second time point in comparison with those with ≥5 CTCs: PFS: 6.9 vs. 2.4 mo., *p* = 0.012 OS: 8.9 vs. 3.9 mo., *p* = 0.003
Xu et al. [31]	Before the 1st, 2nd, and 3rd cycle	66	Chemo	CellSearch	A statistically significant difference between CTC value, a modification after 2 courses and PD was observed (*p* = 0.028)
Muinelo-Romay et al. [74]	Before the 1st, 2nd, and 5th cycles of chemo	43	Chemo	CellSearch	Pts with ≥2 CTCs at 2nd cycle had superior rates of PD and poorer PFS (4.2 mo. vs. 8.5 mo., *p* = 0.016)
Gorges et al. [95]	At baseline and every 12 weeks	50	Chemo	GILUPI CellCollector	Treatment response during therapy was associated with significant decreases in CTC counts (*p* = 0.001)
J Li et al. [83]	At baseline, before 2 and 4 cycle of chemo or after 1 and 2 months of targeted therapy	100	Chemo Targeted therapy (anti-EGFR)	Negative enrichment of immune magnetic beads and FISH	The changes in CTC levels before and after the 2nd cycle of chemo were correlated with the DCR: 38% in CTC-positive pts vs. 62% in CTC-negative pts (*p* < 0.001)
Tamminga et al. [92]	At baseline	86	Chemo (n = 52) TKI (n = 34)	CellSearch	Pts with CTCs vs. pts without CTCs: RR: (a) TKI: 25% vs. 73% (*p* = 0.02) (b) chemo: 35% vs. 51% (*p* = 0.05) PFS: 3.3 mo. vs. 8 mo., *p* = 0.01 OS: 5.2 mo. vs. 12.1 mo., *p* = 0.03
Wang et al. [94]	At baseline, at cycle 1 and cycle 4, and at PD	150	Chemo	CellSearch	Pts with persistent negative CTCs had longer PFS (5.7 mo. vs. 3.0 mo.) and OS (13.1 mo. vs. 5.6 mo.) compared with pts with persistent positive CTCs, which was not impacted by chemotherapy. Chemotherapy decreased CTC from 36.0% to 13.7%
Zhou et al. [85]	Before the 1st and the 3rd cycle	59	Chemo	CellSearch	No correlation between the change of CTC counts and response to chemotherapy was observed
Hirose et al. [76]	At baseline	33	Chemo	CellSearch	No differences in response rates to cytotoxic chemotherapy between CTC-positive patients and CTC-negative patients.

### 4.2. Predictive Role in Oncogene-Addicted NSCLC during Targeted Therapy

The predictive value of CTCs has been investigated in oncogene-addicted NSCLC patients receiving targeted therapy, especially in those with EGFR mutations and ALK rearrangements.

A phase II trial investigated the role of CTCs in 41 unselected patients with relapsed or refractory metastatic NSCLC treated with pertuzumab and erlotinib. CTC count was analysed using CellSearch at baseline and during the course of treatment. CTC value decreased in a statistically significant manner from day 56 of treatment, with a significant prolongation of longer PFS (*p* = 0.050) [97]. In the same line, another phase II trial using flow cytometry evaluated the correlation between the change of CTC values and response to first-generation *EGFR* TKIs in 66 advanced, *EGFR*-mutant NSCLC patients. No significant differences were observed in CTC counts between baseline and after treatment in patients with progressive disease (n = 26), whereas the CTC values were significant reduced during treatment in 40 patients with clinical benefit (*p* = 0.009) [98]. Moreover, PFS and 1-year and 3-year OS rates were longer for patients with low CTC counts than those with high CTC counts [98].

Finally, Breitenbuecher et al. evaluated the predictive role of CTCs in a small sample of treatment-naive patients with advanced *EGFR* DelEx19-mutated NSCLC [99]. Fifty percent of patients with *EGFR*-mutant positive CTC samples at baseline became negative within a median of 34 days of treatment, and CTC negativity lasted for a median of 109 days. The negativization of *EGFR*-mutant CTCs was associated with radiological and clinical response to treatment. Moreover, an exploratory analysis showed a significant statistical correlation between the duration of CTC negativity and the time to treatment failure [99].

In a similar way, a limited number of studies reported the ability to detect *ALK*-rearrangement in CTCs from *ALK*-rearranged NSCLC patients [100,101]. Interestingly, the correlation between dynamic modification of CTC counts during treatment and treatment response or disease progression in patients with *ALK*-rearranged NSCLC was confirmed in more studies [102,103]. The results of some studies which investigated the predictive value of CTCs in this setting of patients are shown in Table 4.

### 4.3. Identification of Mechanisms of Resistance in Oncogene Addicted NSCLC during Treatment

In an attempt to identify drug resistance mechanisms, investigators explored the impact of molecular profiling of CTCs in NSCLC patients.

Maheswaran et al. reported a correlation between the emergence of T790M mutations in CTCs and acquisition of drug resistance in patients affected by advanced NSCLC treated with gefitinib [109]. Recently, a Japanese research group investigated the role of BIM-γ mRNA expression on CTCs in 30 *EGFR*-mutated NSCLC patients treated with osimertinib [107]. No significant different was found in terms of objective response rate (ORR) or PFS for osimertinib between CTC-positive patients with ≥10 CTCs per 7.5 mL and those with <10 CTCs per 7.5 mL. The ORR to osimertinib was inferior in patients with high expression of BIM-γ mRNA compared with those with low BIM-γ mRNA expression (26.6% vs. 73.3%, respectively; *p* = 0.011). However, there was no significant difference in terms of disease control rate (DCR), PFS, and OS between the groups. The authors concluded that BIM-γ mRNA overexpression in CTCs in this subset of patients could be a potential biomarker for inadequate response to osimertinib [107].

Moreover, a Greek study group studied changes in gene expression in CTC-enriched fractions of 30 *EGFR*-mutant NSCLC patients under osimertinib [104]. RNA-based molecular characterization of CTCs was performed before, after 1 cycle of treatment, and at progression of disease. They observed a dynamic role of EMT as a resistance mechanism during osimertinib treatment in these patients. Indeed, at PD, they compared the different CTC profiles, observing significant correlations between the epithelial CTC profile and *ALDH-1* (*p* = 0.043), and the mesenchymal/EMT profile and *ALDH-1* (*p* = 0.014). Interestingly, there was also a significant increase in the expression levels of PD-L1 in CTCs between baseline and disease progression (*p* = 0.016). Moreover, a correlation was found between PD-L1 status and the mesenchymal subtype (*p* = 0.036), and more precisely with *VIM* overexpression (*p* = 0.011) at PD. Therefore, based on these observations, the authors suggested a potential role of ICIs in this subpopulation of patients [104].

Recently, Chen et al. investigated drug-resistant gene function and dynamic change of CTCs in 20 *EGFR*-positive NSCLC patients before and after treatment with osimertinib. They found that osimertinib increased the expression of HER-2 on the cell surface of NSCLC cell lines as a drug resistance mechanism [110].

More recently, scientific interest has been focused on CTC profiling in patients with *ALK*-rearranged NSCLC. In a study by Pailler et al., the presence of resistance mutations in CTCs of 17 *ALK*-rearranged NSCLC patients progressing on Crizotinib (n = 14) or Lorlatinib (n = 3) was investigated. In spite of the limitations of this study, they highlighted the high genetic heterogeneity in both *ALK*-independent and *ALK*-dependent pathways, and potential clinical importance of CTCs to identify therapeutic resistance mutations in this population [103].

As a continuation of this study [61], Oulhen et al. [111] analysed 6 of these 17 *ALK*-rearranged patients with ≥10 CTCs/20 mL blood at resistance to 1st and 3rd ALK-TKIs. They described the copy number alteration (CNA) heterogeneity based on epithelial and/or ALK expression features. Phenotypic analysis showed similar CTC features for each patient that were either ALK^−^/CK^+^ or ALK^+^/CK^−^. CNA analysis performed on 43 CTC samples showed a significant CNA heterogeneity and high CIN at treatment resistance. *RTK/RAS* was found to be the predominant mutated ALK-independent pathway revealed by CTCs, followed by multiple gains in *EGFR*, *FGFR1*, *ERBB2*, and *NTRK*. Based on these results, authors suggested a potential utility of CNA evaluation on CTCs for better defining the different resistance mechanisms to ALK-TKIs, and identifying potential alternative targets bypassing these drug resistance pathways [111].

### 4.4. PD-L1 Expression in CTCs during Treatment with Immune Checkpoint Inhibitors

Programmed death ligand 1 (PD-L1) represents the tumour tissue biomarker in order to predict the responsivity to treatment with ICIs in NSCLC. The reliability to predict the response to ICIs of PD-L1 can be related to its variable expression within and between tumour sites and in different time points of tumour evolution [112]. Given that, the use of CTCs may overcome tumour spatiotemporal heterogeneity, and predict responses to ICIs in order to investigate PD-L1 expression on them [113]. In a study by Nicolazzo et al., 24 advanced NSCLC patients treated with nivolumab have been evaluated for PD-L1 expressing CTCs at three predefined timepoints (after 1, 3, and 6 months after starting treatment). Interestingly, the persistence of PD-L1-positive CTCs after 6 months of treatment was correlated with a lack of treatment response [114]. Similarly, Guibert et al. confirmed the prognostic role of PD-L1-positive CTCs in 96 advanced NSCLC patients treated with nivolumab using the ISET method [67]. However, PD-L1-positive CTCs have not shown a clear predictive effect. Moreover, as reported in most studies, they confirmed the lack of correlation between PD-L1 positivity in tumour tissue and PD-L1 positivity on CTCs [67].

Only one study suggested that PD-L1 expression on CTCs could be used as a valid alternative to tumour tissue PD-L1 evaluation for monitoring ICI treatment response, based on their results that demonstrated a correlation of PD-L1 status in CTCs and tumour tissue [86]. Furthermore, Raimondi et al. showed that the presence of PD-L1-positive CTCs isolated from a population of NSCLC patients were correlated with a partial EMT phenotype, suggesting that coexistence of both of these markers represents a possible immune escape mechanism [115]. To the contrary, Kulasinghe et al. reported no association between PFS and presence of PD-L1-positive CTCs in advanced NSCLC patients [91]. Spiliotaki et al. explored PD-L1 and Ki-67 in CTCs of 47 advanced NSCLC patients receiving pembrolizumab [116]. Blood samples were analysed at baseline, post-first cycle, post-third, and primary resistance (PMR). Patients in PD presented a higher number of total and PD-L1-low CTCs after the first cycle compared to patients with clinical benefit (83% vs. 37% and 67% vs. 8%, respectively). Moreover, patients with a reduction of total and PD-L1-low CTCs after the first cycle of pembrolizumab presented a longer PFS. To the contrary, PD-L1-positive patients with a Ki67 index > 30% on baseline presented shorter PFS and OS compared to those harbouring Ki67 ≤ 30% [116].

A summary of the recent studies evaluating PD-L1 expression on CTCs in advanced NSCLC patients is reported in Table 5.

## 5. Future Perspectives: CTCs vs. cfDNA

One of the most interesting fields in CTC research is the possibility to study their molecular profiling in order to identify biomarkers predictive of response to targeted therapy.

Indeed, several studies analysed the predictive role of some actionable genomic mutations or rearrangements, such as *ALK*, *EGFR*, and *ROS-1* in CTCs of metastatic NSCLC patients, trying to identify drug resistance mechanisms and potentially leading to the development of new target-based agents [61,103,107,109]. However, to date, the field of molecular characterization of NSCLC is dominated by the use of circulating tumour DNA (cfDNA) that is already commonly integrated into clinical practice of NSCLC management, differently from CTCs. This is due to the fact that cfDNA analysis is easier and less expensive compared to CTC analysis [120]. In addition, the number of CTCs isolated in patients with NSCLC is too limited to consider molecular analysis.

However, CTC evaluation has some advantages over cfDNA, and enables one to obtain different information. First of all, CTC analysis allows the detection of protein expression, as PD-L1 can be a potential predictive marker in NSCLC patients treated with ICIs [67,114]. Second, on CTCs, it is possible to analyse morphological features. Finally, molecular profiling of CTCs depicts better tumour heterogeneity in spatiotemporal terms compared to cfDNA [121]. Recent studies have also demonstrated the significant role of CTCs in ex vivo models, including mouse xenografts [17,122], investigating the process of metastatization and of drug resistance mechanisms, leading to new therapeutic opportunities.

In the near future, we expect interesting progresses in this research field because of a progressive improvement of platforms for single-cell analysis of CTCs. Moreover, an alliance between CTC and cfDNA analysis will improve the clinical application of liquid biopsy, leading to an earlier identification of nonresponders who may benefit from an early switch to a more effective alternative treatment.

## 6. Conclusions

Despite a significant heterogeneity in the studies investigating the role of CTCs in terms of patient characteristics, CTC cut-off value, and methods of CTC analysis, the results of the majority of studies seem to be concordant regarding the correlation between CTC count and survival endpoints in advanced NSCLC. Moreover, several studies have confirmed the predictive role of CTCs during different types of therapies in terms of treatment response or drug resistance. Nevertheless, while the use of CTCs in NSCLC seems to be promising, their use was not implemented in routine clinical practice, and some important obstacles need to be overcome.

Development of well-standardized platforms with higher sensitivity and specificity for the detection and molecular analysis of CTCs, and subsequent validation of their role in large cohorts of patients, are unquestionably necessary. Moreover, large-scale prospective randomized trials with homogeneous patient features and robust endpoints should be conducted for validation of CTCs in NSCLC patients and their subsequent implementation in clinical practice. Furthermore, based on limited data regarding the prognostic and predictive role of CTCs in specific molecular subgroups of NSCLC patients eligible for targeted therapies or in patients treated with immune checkpoint inhibitors, additional research in these groups is absolutely necessary.

In conclusion, in the era of precision oncology, CTC analysis could be a promising tool for prognosis, selection of patients most likely to benefit from personalized treatment, and improving of our knowledge of the molecular evolution of this disease.

## Figures and Tables

**Figure 1 ijms-24-16085-f001:**
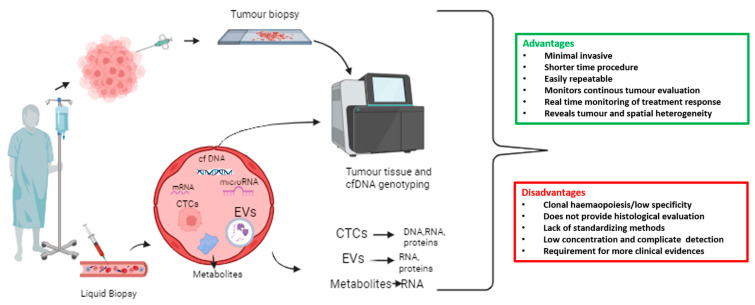
Overview of liquid biopsy approach with advantages and disadvantages. A wide range of biomarker sources are included in the liquid biopsy, such as circulating tumour cells (CTCs), extracellular vesicles (EVs), messenger RNAs (mRNAs), microRNAs, cell-free DNA (cfDNA), and tumour-derived metabolites that are present in bodily fluids such as blood.

**Table 1 ijms-24-16085-t001:** Current technologies for CTC enrichment and detection used in NSCLC. Abbreviations: EpCAM, epithelial cell adhesion molecule.

	Technology	Approach	Advantages	Disadvantages
EpCAM-based methods	CellSearch [27,28]	Anti-EpCAM antibodies	FDA-cleared in advanced breast, colon, prostate tumours	CTCs undergoing EMT may loss EpCAM expression and may not be detected
			Semiautomated	Identifies cell with high expression of EpCAM
	GILUPI Cell-collector [29]	Captures CTCs directly into peripheral vein	In vivo, fast processing of high peripheral blood volumes	More invasive for the patient than a simple blood draw
			The functional domain of the Cell-collector is coated with anti-EpCAM antibodies	Recovers only EpCAM+ CTCs
	Adnatest [30]	Anti-EpCAM antibodies	High purity rates	Recovers only EpCAM+ CTCs
				Cells are lysed
				Potential issues with antibody specificity
Marker-independent methods	Parsortix [31]	Microfluidic based on physical properties	Recovery of viable CTCs	Difficulty of removing the leukocytes of similar size to CTCs
			Ability to detect CTC clusters	
			Detects EpCAM negative CTCs	
	CanPatrol [32]	mRNA in situ analysis	Detects CTCs regardless of markers expression	Blood characteristics may negatively impact on analysis
	ISET [33]	Size-based filtration exclusion	Quick	Loss of CTCs when size is similar to white blood cells
			Nonselective	The identification of CTCs in the filter is complex and time-consuming

**Table 4 ijms-24-16085-t004:** Selective recent studies investigating the predictive value of CTCs in NSCLC patients treated with TKI. Abbreviations: NSCLC, non-small cell lung cancer; CTC, circulating cancer cell; TKI, tyrosine kinase inhibitor; ISET, isolation by size of tumour cells; CK, cytokeratin; EMT, epithelial–mesenchymal transition; VIM, vimentin; PD-L1, programmed cell death ligand 1; PFS, progression-free survival; PFS 12 m: one-year progression-free survival; RR, response rate; *EGFR*, epidermal growth factor receptor; *ALK*, anaplastic lymphoma kinase; OS, overall survival; CNG, copy number gain; pts, patients.

Study	Analysis Time-Point	Patients (n)	Type of Targeted Treatment	Detection Method	Predictive Relevance
Ntzifa et al. [104]	At baseline, after 1 month, and at PD	30	Osimentinib	Parsortix ISET	There was a strong positive correlation of VIM expression with PIM-1 expression at baseline and increased PD-L1 expression levels at PD. The high prevalence of VIM-positive CTCs suggests a dynamic role of EMT during osimertinib treatment
Kallergi et al. [105]	At baseline, post-cycle 1, and at the end of treatment	47	Osimertinib	ISET	The decrease in CTCs occurring early during osimertinib treatment is predictive of better outcome
Pantazaka et al. [106]	At baseline, post-cycle 1, and at the end of treatment	42	Osimertinib	ISET	Significant correlation between PD-L1 and pS6 phenotypes at all time points was found. Survival analysis revealed that CK + pS6+ and CKlowpS6+ phenotypes after 1st cycle were related to significantly decreased PFS12m (*p* = 0.003) and PFS (*p* = 0.021), respectively
Isobe et al. [107]	At baseline	30	Osimertinib	ClearCell FX	RR to osimertinib was worse in pts with high than in those with low BIM-γ mRNA expression (26.6% vs. 73.3%, respectively; *p* = 0.011). PFS did not significantly differ between groups (*p* = 0.13)
Tamminga et al. [92]	At baseline	86	TKI therapy (n = 34)	CellSearch	Pts with CTCs vs. pts without CTCs: RR: TKI: 25% vs. 73% (*p* = 0.02) PFS: 3.3 mo. vs. 8 mo., *p* = 0.01 OS: 5.2 mo. vs. 12.1 mo., *p* = 0.03
Jiang et al. [108]	At baseline, 1 month after treatment, and every 2 months	232	EGFR-TKIs	CytoploRare	Pts with baseline low CTCs vs. high CTCs: PFS: 412 vs. 267 days; HR = 0.48; *p* < 0.001 OS: 836 vs. 583 days; HR = 0.52; *p* = 0.002 Pts with *EGFR*19delmut and low CTCs in comparison with high CTCs have prolonged PFS (HR = 0.51, *p* = 0.014) and OS (HR = 0.52, *p* = 0.036) Pts with *EGFR* L858R) and low CTCs in comparison with high CTCs have prolonged PFS (HR = 0.5, *p* = 0.023) and OS (HR = 0.43, *p* = 0.0007)
Pailler et al. [103]	At baseline and at an early time-point (2 months)	39	Crizotinib	ISET	The dynamic change of CTC with *ALK-CNG* was the strongest factor associated with PFS (14 mo. for pts with decreased *ALK-CNG* vs. 6.1 mo. for pts with stable or increased *ALK-CNG*, *p* = 0.025). No correlation with OS
Pailler et al. [61]	At PD	17	Crizotinib Lorlatinib	ISET CellSearch RosetteSep	Multiple mutations in various genes (*EGFR*, *KRAS*, *BRAF* genes and *TP53* pathways) in *ALK*-independent pathways were predominantly identified in CTCs of crizotinib-resistant patients. In one lorlatinib-resistant patient, 2 single CTCs out of 12 harboured *ALK* compound mutations. CTC-1 harboured the *ALK*^G1202R/F1174C^ compound mutation, virtually similar to *ALK*^G1202R/F1174L^ present in the corresponding tumour biopsy. CTC-10 harboured a second *ALK*^G1202R/T1151M^ compound mutation not detected in the tumour biopsy

**Table 5 ijms-24-16085-t005:** Selective studies investigating the predictive role of PD-L1-positive CTCs in advanced NSCLC patients in treatment with ICIs. Abbreviations: NSCLC, non-small cell lung cancer; CTC, circulating cancer cell; ICI, immune checkpoint inhibitor; PD-L1, programmed cell death ligand 1; PD, progressive disease; ISET, isolation by size of tumour cells; PFS, progression-free survival; OS, overall survival; pts, patients; Nivo, nivolumab; EMT, epithelial–mesenchymal transition; MTV, metabolic tumour volume; MCA, automated microcavity array; PR, partial response; NR, not reached; Pembro, pembrolizumab.

Study	Analysis Time-Point	Patients (n)	Therapeutic Approach	Detection Method	Predictive Relevance
Nicolazzo et al. [114]	At baseline and at 3 and 6 months	24	Nivo	CellSearch	At 6 months of treatment, patients with PD-L1-negative CTCs all obtained a clinical benefit, while patients with PD-L1(+) CTCs all experienced PD
Guibert et al. [67]	At baseline and at PD	96	Nivo	ISET	The presence of PD-L1 + CTCs at baseline was not significantly correlated with PFS (*p* = 0.55) or OS (*p* = 0.89), but a higher baseline PD-L1+ CTC number (≥1%) was observed in the “non-responders” group (PFS < 6 mo.) (*p* = 0.04), and PD-L1 + CTC were seen in all patients at PD
Raimondi et al. [115]	At baseline	15	Nivo	ScreenCell Cyto	Coexpression of PD-L1 and EMT markers might represent a possible molecular background for immune escape in pts with NSCLC under treatment with Nivo
Spiliotaki et al. [116]	At baseline, post-1st cycle, post-3rd cycle, and at primary resistance	47	Pembro	Ficoll density gradient centrifugation	PD-L1^+^ Pts with Ki67^+^ CTCs > 30% before treatment had a shorter PFS compared to those with a Ki67 ≤ 30% (2 mo. vs. NR, respectively, *p* < 0.001); OS was shorter in PD-L1^+^ patients harbouring Ki67^+^ CTCs compared to those not presenting (median OS: 11.8 mo. vs. 33.1 mo., respectively; *p* = 0.035). In sequential samples of patients with a durable benefit, a low Ki67 index was observed
Castello et al. [117]	At baseline and 8 weeks after ICI initiation	35	ICIs	ISET	The combination of mean CTC and median MTV at 8 weeks was associated with PFS (*p* < 0.001) and OS (*p* = 0.024). CTC number is modulated by previous treatments and correlates with metabolic response during ICI
Ikeda et al. [118]	At the baseline and weeks 4, 8, 12, and 24	45	Nivo	MCA system	The PD-L1 positivity rate in CTCs at week 8 or immediate increase in the number of CTCs prior to achieving PR and the decrease thereafter were significantly correlated with response to Nivo
Zhou et al. [66]	At baseline	139	ICIs	Cyttel method	No correlation between PD-L1 expression to CTCs and tumour tissues. Pts with PD-L1 detected on CTCs or tissue achieved significantly prolonged PFS compared to those without PD-L1 (5.6 mo. vs. 1.4 mo., log-rank *p* = 0.032)
Janning et al. [119]	At baseline, during treatment, and at PD	127	ICIs	Parsortix system	The percentage of PD-L1^+^ CTCs did not correlate with the percentage of PD-L1^+^ in biopsies (*p* =0.179). Upon PD, all pts showed an increase in PD-L1^+^ CTCs, while no change or a decrease in PD-L1^+^ CTCs was observed in responding patients (*p* = 0.001).

## Data Availability

Data sharing not applicable.

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
