# Peer review of "Circulating Tumour Cells: Detection and Application in Advanced Non-Small Cell Lung Cancer"

_ijms, 2023, doi:10.3390/ijms242216085_

Round 1

Reviewer 1 Report

Comments and Suggestions for Authors

The authors provide a review looking at CTC in NSCLC.

This is also important information given that most of the clinical applications  to date have been on cfDNA or ctDNA through the use of NGS, and applications at point of diagnosis, on treatment progression or on minimal residual disease and there is a dearth of information regarding the use of CTC. The limitation for CTC (as acknowledged by the authors) is that it is quite rare, requires more specialised methods to isolate and detect and is also prone to tumor hetereogeneity.

The authors delve into the methodology: 2 main categories of CTC search (either EP-cam approach or independent of biomarker). It may be useful for authors to also provide the search term and methodology used in the literature review to ensure that is is comprehensive or the most updated. There could also be emerging technologies that have yet to be published or clinically validated.

For PD-L1 testing, tissue testing via IHC may still remain gold standard and more information may be needed on how liquid biopsy may be utilised (?) While this may be possible, it remains to be seen if it is used in a clinical setting, futhermore, there have been differences in interobserver varability with PD-L1 and also different scoring used in different companion diagnostics of IO

The authors then proceed to elaborate on the use of CTC in the various stages of the patient journey, the type of treatment (EGFR-TKI or ICI) and the outcome observed.

The authors may elaborate further on the findings of this various studies, as often it is hard to come to a generalisation due to how each of the study design may be different (for instance, is the differences in observed outcomes attributed to the difference in patient population, time point of sampling of technological limitation in not being able to detect certain CTCs). This would be useful point to add to the review. In addition, to showing the challenges that have been identified through literature review, what are the authors' thoughts on how best to address these?

Suggestion: To also include a overview with figure of the other liquid biopsies method and its application along the NSCLC patient journey. This can give a top-line view, while tables remain informative, it does not give the descriptive summary

Author Response

Reviewer 1

The authors provide a review looking at CTC in NSCLC.

This is also important information given that most of the clinical applications  to date have been on cfDNA or ctDNA through the use of NGS, and applications at point of diagnosis, on treatment progression or on minimal residual disease and there is a dearth of information regarding the use of CTC. The limitation for CTC (as acknowledged by the authors) is that it is quite rare, requires more specialised methods to isolate and detect and is also prone to tumor hetereogeneity.

The authors delve into the methodology: 2 main categories of CTC search (either EP-cam approach or independent of biomarker). It may be useful for authors to also provide the search term and methodology used in the literature review to ensure that is is comprehensive or the most updated. There could also be emerging technologies that have yet to be published or clinically validated.

Reply: We provided literature review methods as requested (lines 55-56). In addition, we included the description of an emerging technology for lung metastasis initiating cells detection in the blood which combines Raman spectroscopy and machine learning (lines 252-260), as suggested.

For PD-L1 testing, tissue testing via IHC may still remain gold standard and more information may be needed on how liquid biopsy may be utilised (?) While this may be possible, it remains to be seen if it is used in a clinical setting, futhermore, there have been differences in interobserver varability with PD-L1 and also different scoring used in different companion diagnostics of IO.

Reply: Immunohistochemistry (IHC) is not suitable for liquid biopsy, and PD-L1 determination on CTCs requires other approaches. We cited some studies in which immunofluorescence, CellSearch and other techniques (lines 330-334).

The authors then proceed to elaborate on the use of CTC in the various stages of the patient journey, the type of treatment (EGFR-TKI or ICI) and the outcome observed.

The authors may elaborate further on the findings of this various studies, as often it is hard to come to a generalisation due to how each of the study design may be different (for instance, is the differences in observed outcomes attributed to the difference in patient population, time point of sampling of technological limitation in not being able to detect certain CTCs). This would be useful point to add to the review. In addition, to showing the challenges that have been identified through literature review, what are the authors' thoughts on how best to address these?

Reply: We thank the reviewer for the comment. In this context, we think that the lack of standardization could be an issue. We remarked this point in the discussion (lines 730-732).

Suggestion: To also include a overview with figure of the other liquid biopsies method and its application along the NSCLC patient journey. This can give a top-line view, while tables remain informative, it does not give the descriptive summary

Reply: We added a Figure 1 describing different liquid biopsy methods as suggested.

Reviewer 2 Report

Comments and Suggestions for Authors

Article titled ‘Circulating tumour cells: detection and application in advanced Non-Small Cell Lung Cancer’

is interesting and I read it with interest.

Below are a few comments on the manuscript.

1. In vivo, in vitro, in situ, ex vivo should be in italics.

2. In Figure 2: perhaps it would be worth adding in CONS the short survival of these cells?

3. Throughout the article: if the authors write about genes their names should be italicized.

4. Subsection 2.2.2. perhaps on/in CTCs should be added to the title?

5. section 4-his title: Predictive role and clinical implications in advanced NSCLC. But the role of what?

6. Throughout the article: if the authors write about mutations in EGFR in lung cancer, the name should be in italics to indicate that the gene, not the protein, is being referred to. This is important because the diagnosis of EGFR is done at the level of DNA-gene sequence.

7. If the authors write about changes in ALK or ROS1, it should be distinguished and clearly indicated whether the authors mean mutations or rearrangements.

Author Response

Reviewer 2:

Comments and Suggestions for Authors

Article titled ‘Circulating tumour cells: detection and application in advanced Non-Small Cell Lung Cancer’

is interesting and I read it with interest. Below are a few comments on the manuscript.

In vivo, in vitro, in situ, ex vivo should be in italics.

Reply: We modified the text as suggested.

In Figure 2: perhaps it would be worth adding in CONS the short survival of these cells?

Reply: as suggested by other reviewers we removed this figure. However, we reported some details on the short survival of both single CTCs and CTC clusters in lines 81-83.

Throughout the article: if the authors write about genes their names should be italicized.

Reply: We modified the text as suggested.

Subsection 2.2.2. perhaps on/in CTCs should be added to the title?

Reply: We modified the title as suggested

section 4-his title: Predictive role and clinical implications in advanced NSCLC. But the role of what?

Reply: We modified the title as suggested

Throughout the article: if the authors write about mutations in EGFR in lung cancer, the name should be in italics to indicate that the gene, not the protein, is being referred to. This is important because the diagnosis of EGFR is done at the level of DNA-gene sequence.

Reply: We modified the title as suggested

If the authors write about changes in ALK or ROS1, it should be distinguished and clearly indicated whether the authors mean mutations or rearrangements.

Reply: We specified better this difference in the text.

Reviewer 3 Report

Comments and Suggestions for Authors

Andrikou and co-authors reviewed the literature on CTCs for diagnosing and monitoring NSCLC. There was good coverage of the literature, and the text is well-written. However, there are some addressable details to improve the manuscript. 

Main points

  1. Figure 1 is unecessary and may become a couple of sentences in the text. Alternatively, authors could provide a figure with a summary of the review (types of CTCs, detection methods, outlook of diagnostics and disease monitoring) in the end of the text;
  2. The introduction must provide an outlook of approaches to liquid biopsy (cell-free nucleic acids, exosomes, and so on) and why the focus on CTCs (part of this text is on topic 5 - future perspectives);
  3. Authors should give more details on type of CTCs (single-cells vs. clusters) due the correlation to disease progression; 
  4. Tables needs adjustments do take less space on each page;
  5. Authors should provide details on the criteria for choosing the references to each table;

Minor points

Line 15 (page 1): "...most studied approaches to liquid biopsies";

Lines 17-20 (page 1): The molecular resolution of liquid biopsy could be here;

Line 48 (page 2): no need for "In contrat to tissue biopsy";

Line 25 (page 7): "...and CTC counts".

Line 27 (page 7): Table 2 (and others): suggestions for headings: "CTC biomarkers", "Analysis time-point", "Patients (n)", "detection method", "Thereupic approach", "Main findings";

Line 32 (page 14): "Breitenbuecher et al. evaluated";

Author Response

Reviewer 3:

Comments and Suggestions for Authors

Andrikou and co-authors reviewed the literature on CTCs for diagnosing and monitoring NSCLC. There was good coverage of the literature, and the text is well-written. However, there are some addressable details to improve the manuscript.

Main points

Figure 1 is unecessary and may become a couple of sentences in the text. Alternatively, authors could provide a figure with a summary of the review (types of CTCs, detection methods, outlook of diagnostics and disease monitoring) in the end of the text;

Reply:  We agree with the reviewer, and as suggested we removed Figure 1.

The introduction must provide an outlook of approaches to liquid biopsy (cell-free nucleic acids, exosomes, and so on) and why the focus on CTCs (part of this text is on topic 5 - future perspectives);

Reply:  We added in the main text a new figure 1, which provides an outlook of the liquid biopsy approaches (Figure 1 and lines 61-75).

Authors should give more details on type of CTCs (single-cells vs. clusters) due the correlation to disease progression;

Reply: in accordance with the comment, we added an explanation of this point (lines 77-91).

Tables needs adjustments do take less space on each page;

Reply: we adjusted the tables.

Authors should provide details on the criteria for choosing the references to each table;

Minor points

Line 15 (page 1): "...most studied approaches to liquid biopsies";

Lines 17-20 (page 1): The molecular resolution of liquid biopsy could be here;

Line 48 (page 2): no need for "In contrat to tissue biopsy";

Line 25 (page 7): "...and CTC counts".

Line 27 (page 7): Table 2 (and others): suggestions for headings: "CTC biomarkers", "Analysis time-point", "Patients (n)", "detection method", "Thereupic approach", "Main findings";

Line 32 (page 14): "Breitenbuecher et al. evaluated";

Replay: We reported the suggested changes in the text and tables